# Development and Evaluation of a Novel HNB Based Isothermal Amplification Assay for Fast Detection of Pyrimethamine Resistance (S108N) in *Plasmodium falciparum*

**DOI:** 10.3390/ijerph16091635

**Published:** 2019-05-10

**Authors:** Madhvi Chahar, Anup Anvikar, Neena Valecha

**Affiliations:** Division of Epidemiology and Clinical Research, National Institute of Malaria Research, Sector-8 Dwarka, New Delhi 110077, India; anvikar@gmail.com (A.A.); neenavalecha@gmail.com (N.V.)

**Keywords:** *Plasmodium falciparum*, malaria, pyrimethamine, resistance, *Pfdhfr* gene, LAMP

## Abstract

Sulphadoxine and pyrimethamine (SP) have been used as long-acting partner antimalarial drugs in artemisinin combination therapy (ACT) for falciparum malaria. The emergence and increasing spread of SP resistance in malaria-endemic areas have become a challenge for the control of malaria. Therefore, regular monitoring of the mutation status of partner drugs is important for the better management of drug policy. There are limitations with traditional molecular methods and there is an urgent need for an easy method for diagnosis of drug resistance. In this study we have introduced and developed a novel single nucleotide polymorphism loop-mediated isothermal amplification (SNP–LAMP) approach based on a hydroxynaphthol blue (HNB) indicator for the easier and quicker detection of pyrimethamine resistance in *Plasmodium falciparum* malaria. To implement this novel approach, many sets of LAMP primers were designed and tested. Finally, one set of forward inner primer M1 (FIPM1) of LAMP primer was selected that specifically distinguishes pyrimethamine-resistant *P. falciparum* malaria. The LAMP reactions were optimized at 60–66 °C for 45 min. High sensitivity (7 parasites/µL) was observed with 10^−4^ fold dilutions (<2 ng DNA) of genomic DNA. Moreover, this approach has the potential to be applied even in laboratories unfamiliar with PCR or other molecular methods, and in future, this can be helpful for the better management of anti-malarial policy.

## 1. Introduction

*Plasmodium falciparum* malaria remains one of the major health concerns, with a large number of the world’s population at risk from the disease. One of the greatest challenges for malaria treatment is the emergence and spread of drug resistance in *P. falciparum*, resulting in treatment failure and poor clinical outcomes, highlighting the need to develop rapid and accurate field-based assays for antimalarial resistance detection, which are decisive in choosing drugs when starting therapy [1]. Sulphadoxine and pyrimethamine (SP) have been used as a choice of combination drugs for falciparum malaria [2,3,4]. The efficiency of pyrimethamine and sulphadoxine as partner drugs for falciparum malaria treatment could be predicted by the analysis of *P. falciparum* target genes. Pyrimethamine and sulphadoxine inhibits the dihydrofolate reductase (*Pfdhfr*) and dihydropteroate synthase (*Pfdhps*) biosynthesis pathways in the parasite. Moreover, due to point mutations in the *Pfdhfr* and *Pfdhps* genes the drug could not bind to the active site of the enzyme and result in resistance for pyrimethamine and sulphadoxine [5]. The key mutation in dihydrofolate reductase (*Pfdhfr*) S108N developed pyrimethamine (PYR) resistance in *P. falciparum*, and additive mutations in *Pfdhfr* N51I and C59R also conferred higher levels of resistance [6,7,8,9,10]. Recent reports suggest that there is a significant increase in PYR resistance due to drug pressure and malaria transmission, which could be a cause of treatment failure in the near future [6].

So far, the existing molecular detection methods for antimalarial resistance are time-consuming, tedious, and require trained personnel and sophisticated equipment [1]. Moreover, their use is limited and challenging to implement in resource-limited settings. Loop-mediated isothermal amplification (LAMP) is an easy and convenient technique that can amplify nucleic acids with high specificity, sensitivity, efficiency, and rapidity under isothermal conditions [11]. This technique is characterized by the use of a DNA polymerase with strand displacement activity and a set of two specifically designed primers, namely inner primers (forward inner primer (FIP) and backward inner primer (BIP) and outer primers (F3 and B3) [12,13]. The amplification efficiency of the LAMP assay is extremely high based on the isothermal reaction. Therefore, the LAMP assay has the advantages of specificity and rapidity over other nucleic acid amplification techniques [13].

The LAMP approach has opened new horizons for the monitoring of drug resistance and it could be easily applied to national malaria control programs due to its simplicity and low infrastructure costs. The innovation of a LAMP-based point of care test for detecting a resistant genotype of malaria parasites could be an effective approach to facilitate antimalarial resistance detection and tracking in resource-limited countries afflicted with the disease [1]. Moreover, the applications of single nucleotide polymorphism (SNP)–LAMP assays have successfully detected mutations in *P. falciparum* chloroquine resistance transporter gene (*Pfcrt*) for chloroquine resistance [14,15] and *P. falciparum* dihydrofolate reductase gene (*Pfdhfr*) for pyrimethamine resistance [16].

Relative to LAMP, PCR-based molecular techniques are limited in their suitability for field use due to the requirements of expensive equipment and well-trained personnel, even for basic rapid laboratory detection. Therefore, there is an urgent need for an easy approach like LAMP for early detection of antimalarial drug resistance in an endemic population for planning any pilot surveillance study for the control/correct treatment of falciparum malaria.

To that end, we optimized and developed a novel single nucleotide polymorphism isothermal amplification (SNP–LAMP) assay based on a hydroxynaphthol blue (HNB) indicator for the rapid and easy detection of pyrimethamine resistance in *P. falciparum* malaria. This approach can also be useful even in laboratories unfamiliar with PCR and other molecular analysis methods. Additionally, the standardized approach could also be helpful to develop similar protocols for other important drug targets, i.e., *Pfdhps*, K13 associated with sulphadoxine and artemisinin resistance in *P. falciparum* [5,17].

## 2. Materials and Methods

### 2.1. Chemicals

The chemicals used were as follows: hydroxynaphthol blue (HNB) (Sisco Research Laboratories Pvt. Ltd., India), oligonucleotides (Integrated DNA Technologies, Inc. Skokie, Illinois, USA), nuclease free water, Bst DNA polymerase, dUTPs and MgSO_4_, (New England Biolabs, Inc. Ipswich, MA, USA), betaine (Affymetrix Inc-USB, Thermo Fisher Scientific, Waltham, MA, USA), and agarose (GeNeiTM, Banglore, India). 

### 2.2. P. falciparum Samples

Forty *P. falciparum* positive samples containing the *Pfdhfr* (N108) mutation and sixty samples comprising a *Pfdhfr* wild type allele (S108) were obtained from National Institute of Malaria Research (NIMR), New Delhi, India. These samples were previously characterized by PCR–RFLP for mutation (S108N) associated with pyrimethamine resistance in *P. falciparum* and were used to determine the efficacy of the HNB–LAMP assay. The control strains, including a wild-type control (3D7) and mutant control (DD2), were provided by the Malaria Parasite Bank of National Institute of Malaria Research (NIMR), New Delhi, India. Clinical sample collections were approved by the Institutional Ethical Review Committee of the National Institute of Malaria Research (NIMR), New Delhi, India. 

### 2.3. Genomic DNA Extraction and Quantification

Genomic DNA of *P. falciparum* samples were extracted by using the QIAamp DNA Blood Mini/Maxi Kit (Qiagen) as per the manufacturer’s protocol. The optical densities (OD) of the extracted DNA samples were checked by using a spectrophotometer (Nanodrop Technologies). A tenfold serial dilution was used to determine the detected limit of LAMP. The HNB–LAMP sensitivity was determined up to 10^4^-fold dilution (<2 ng/μL DNA).

### 2.4. In-House Design LAMP Primers

Five sets of primers were designed and synthesized for LAMP assay by using Primer Explorer V4 software [11]. The sequences of *P. falciparum* (GenBank accession number XM_001351443) were used to design the target for the most widely reported mutations (S108N) in the *Pfdhfr* gene associated with the molecular detection of PYR resistance. The LAMP primer sets and their sequence alignment used in this study are given in Table 1 and Figure 1.

### 2.5. Specificity of LAMP Primers

To expand the specificity of the LAMP primers, one or two base-pair mismatches were introduced within three nucleotides of the 3′ end of the forward inner primer (FIP) (Figure 1). Thus, five sets of LAMP primers were synthesized by adding different bases at the 3’ end of the FIP and tested to distinguish the *Pfdhfr* S108N mutants from the wild type. To validate the accuracy of the LAMP primers, genomic DNA of the control strain DD2 (mutant control) and 3D7 (wild type control) were tested. The LAMP assay was performed at the given standard optimum conditions; each LAMP product was analyzed by gel electrophoresis on 3.0% agarose gel. In addition, the LAMP amplification reaction was seen by the naked eye, with the HNB color changing from violet to blue if the required product was present; otherwise, it remained violet in the absence of product.

### 2.6. Optimization of LAMP Assay

The LAMP reaction was optimized in a total volume of 10 μL. The reaction reagents were standardized at various concentration ranges for HNB (100–200 μM), Bst DNA polymerase (0.8–2 U), dNTPs (0.1–1.0 mM), primers FIP and BIP (0.5–2.0 μM) and F3 and B3 (0.2–0.8 μM), betaine (0.3–1 M), and MgSO_4_ (0.5–6 mM). The maximum amplification in the LAMP reaction was obtained with a 10 μL reaction volume containing 2.0 U Bst DNA polymerase, 2.5 mM MgSO_4_, 0.6 mM dNTPs, 0.5 M betaine, 120 μM HNB dye, 1.2 μM FIP and BIP, 0.3 μM F3 and B3, and 1.5 μL (< 2 ng/μL) of DNA. The reaction was optimized at 57 to 66 °C temperature and time intervals of 15, 30, 45, 60, 75, 90, and 105 min. The optimum conditions for amplification of the LAMP reaction was observed at 60–66 °C for 45 min with termination at 90 °C for 5 min. To confirm the constancy, each assessment was repeated thrice with three independent replicates.

### 2.7. Corroboration of LAMP Assay via Sequencing

For the endorsement and precision of LAMP results, PYR-resistant samples with the *Pfdhfr* S108N mutation and all the wild types were validated through DNA sequencing. The specific targets of the *Pfdhfr* gene from all samples were amplified through forward (F3) and backward (B3) primers (Table 1) using PCR, and the PCR products were sequenced through a commercially available sequence service of New England Biolabs, Ipswich, MA, USA. 

### 2.8. Statistical Analysis

Statistical analysis was accomplished by using the Statistical Package for the Social Sciences (SPSS) software, version 10. The frequencies and 95% confidence interval (CI) were used for categorical variables. The sensitivity, specificity, and accuracy of LAMP, PCR–RFLP, and gold-standard sequencing reference tests were evaluated by Chi square test or Fisher’s exact test. The level of statistical significance was set at a value of *p* ≤ 0.05. In this study, each test was repeated thrice, leading to three practical modalities.

### 2.9. Ethical Approval

Clinical sample collections were approved by the Institutional Ethics Review Committee (ERC) of the National Institute of Malaria Research (ICMR), New Delhi, India. This study has been reviewed and approved by Ethical Committee of the National Institute of Malaria Research (ECR/NIMR/EC/2016/121).

## 3. Results

The current study reported the development of the HNB–LAMP assay, which was analyzed with the naked eye by observing the blue color for the amplification reaction produced by hydroxynaphthol blue dye (HNB). In this assay, the applied forward inner primers were specifically designed by inserting a mutant target region to specifically discriminate pyrimethamine-resistant mutant samples from wild types (Figure 1). 

### 3.1. Optimization and Specificity of LAMP Primers

Many sets of LAMP primers were designed to detect PYR resistance encoding *Pfdhfr* gene at S108N point mutation. Five sets of primers were used for primary screening for an appropriate set of LAMP primers to differentiate clearly the S108N mutant and wild-type genomic DNA of *P. falciparum*. In-house design primer sequences are given in Table 1. Out of five sets, only one primer set (FIPM1) was qualified for accurate detection of the S108N mutants associated with pyrimethamine-resistant *P. falciparum* (Figure 2 and Figure 3). The specificity of the LAMP primers was enhanced by adding one mismatch to the forward inner primer (FIP), enabling specific amplification at the target position so the primers would specifically amplify the target position S108N (Figure 2). Hydroxynaphthol blue (HNB) dye induced a color change to blue in the positive reaction, and it was easy to visually discern this compared to the violet-colored negative reaction (Figure 2A). This was further reconfirmed through gel electrophoresis by observing the ladder-like pattern in the mutant strain (DD2, positive control) and no band pattern in the wild type (3D7, negative control) (Figure 2B). Our findings indicated that the in-house designed LAMP primer set FIPM1 was found to be appropriate for specifically distinguishing PYR-resistant mutant (S108N) samples of *P. falciparum*.

For the confirmation of LAMP specificity, we checked the cross reactivity with a mixture of heterogeneous parasite populations. Pyrimethamine-resistant parasites (DD2) were mixed with pyrimethamine-sensitive parasites (3D7). DNA samples of mixed parasite populations were mixed with DD2 and 3D7, respectively, and detected by LAMP assay using *Pfdhfr* gene S108N as a reliable marker to detect PYR-resistant *P. falciparum*; the experiment was repeated three times with positive and negative control reference strains. The LAMP assay clearly discriminated the mutant and wild type strains from genetically mixed population. The results were further validated by RFLP and sequencing (data not given).

*Pfdhfr* gene specific in-house designed LAMP primers specified as forward inner primer (FIPM1, M2, M3, M4, and M5) and backward inner primer (BIP) with two external primers, F3 and B3: At the principal stage, the target sequence was documented by six independent sequences (F1c, F2, F3, B1c, B2, and B3), the inner primers (FIP, BIP) were documented by four independent sequences (F1c, F2, B1c, and B2), and target amino acid change was introduced in the 5 sets of forward inner primers (FIPM1, M2, M3, M4, and M5). 

### 3.2. Optimization of HNB–LAMP Assay

The LAMP reaction was performed with one hundred well characterized samples of *P. falciparum*, including the mutant control (DD2) as positive control and wild-type control strain (3D7) for negative control. The reaction was optimized at different temperature ranges from 57 to 66 °C, with a time interval of 15–105 min, and a color change in the reaction was monitored (Figure 4 and Figure 5). The highest intensity of color change was observed in the tube reaction and band pattern in gel electrophoresis at 60–66 °C (Figure 4A,B). The LAMP product was amplified and visualized at time intervals of 15 min; in terms of fast detection, the best amplification was observed at 45 min, which was further confirmed by the gel electrophoresis of the LAMP product (Figure 5A,B). The investigation concluded the best optimal LAMP reaction was observed at 60 °C for 45 min.

### 3.3. Accuracy and Specificity of HNB–LAMP Assay

The optimization of in-house designed LAMP primers is crucial for the precise and specific detection of an HNB–LAMP assay. The selected set of target specific primers clearly distinguished the wild-type (3D7) and mutant (DD2) control strains (Figure 6). The specificity of the reaction was further assessed through the HNB dye by observing the color change in the tubes (Figure 6A). In the reaction the mutant positive samples turned a blue color in the tube while the negative samples remain violet (Figure 6A). The reaction was reconfirmed by observing a ladder-like pattern on agarose gel (Figure 6B). The positive mutant samples showed a ladder-like pattern on 3.0% gel electrophoresis while there were no bands in the negative samples (Figure 6B). Our results indicated that out of five primer sets—namely FIPM1, M2, M3, M4, and M5 (Figure 2 and Table 1)—only one set (FIPM1) was found to be accurate to specifically distinguish the PYR-resistant mutant samples of *P. falciparum* (Figure 3). The primer set FIPM1 was finally selected for further study. The LAMP reactions were validated by gold-standard sequencing, with 100% accuracy and high specificity of the standardized LAMP method (Table 2).

### 3.4. Standardization of LAMP Reaction Components and Sensitivity Detection

The LAMP detection limit was standardized through the serial dilution of the DNA samples of *P. falciparum*. The reaction showed the high sensitivity up to the detection limit of 7 parasites/μL in 10^−4^-fold dilution (Figure 7). The used parasite samples contained initial densities ranging between 0.02 and 1 parasitized cell per 100 red blood cells as determined by microscopy. This LAMP sensitivity was confirmed by HNB-induced color change at different dilutions from 10^−1^ to 10^−6^ and further validated with 3% agarose gel electrophoresis. The lowest detection limit of diluted DNA samples was successfully observed in 10^−4^-fold dilution (<2 ng/μL DNA), which showed high-sensitivity detection with 7 parasites/μL.

The LAMP reaction components were also optimized to get high specificity using genomic DNA of the mutant *P. falciparum* samples, including a positive control strain (DD2) as the template. The positive reaction showed color change with HNB dye (Figure 3A) and showed the ladder-like pattern with 3% gel electrophoresis (Figure 3B), while there was no amplification observed in the wild-type control (3D7) and ddH_2_O control. Finally, a 10μL reaction was standardized that comprised 2.0 U Bst DNA polymerase, 1.0 μL (10X) ThermoPol buffer, 2.5 mM MgSO_4_, 0.6 mM dNTPs, 0.5 M betaine, 120 μM HNB, 1.2 μM each of FIP and BIP, 0.3 μM each of F3 and B3, and 1.5 μL (<2 ng/μL) of target DNA.

### 3.5. LAMP Assay Corroboration and Constancy Test

The LAMP results were corroborated with gold standard sequencing results using the above *P. falciparum* samples. The LAMP results showed 100% similarity with sequence analysis, which confirmed the LAMP specificity to detect and amplify the single nucleotide change within the *Pfdhfr* gene region at the S108N mutation. The LAMP constancy and repeatability were confirmed using 40 mutant (S108) samples and 60 wild-type samples including control strains (Appendix A) from parasite bank of National Institute of Malaria Research, New Delhi, India. Each reaction was done thrice autonomously for the LAMP assay. The all mutant strains were clearly distinguished from wild type strains through the standardized LAMP assay. The analysis was based on HNB visualization and agarose gel electrophoresis (Figure 3 and Figure 6). The sensitivity, specificity, and accuracy of the LAMP assay were statistically significant with PCR–RFLP and gold-standard sequencing (Table 2).

The gold-standard sequencing accomplished a level of agreement with the HNB LAMP and LAMP–gel (100%, kappa (κ) = 1.0) (Table 2). The visual LAMP analysis appeared to be high with a corresponding kappa value of 1.0. The κ statistics (ranging from 1 to −1) measured the detected percentage of agreement from <0 = poor agreement to 1 = perfect agreement. On the basis of the analysis, it is recommended that the developed HNB–LAMP technique had good stability and repeatability to distinguish the PYR-resistant and wild-type samples of *P. falciparum.*

### 3.6. Application of HNB–LAMP in Monitoring Pyrimethamine Resistance

This study demonstrated a novel HNB–LAMP assay for the fast monitoring of pyrimethamine-resistant samples of *P. falciparum* in low-resource settings. This approach was based on an easy-to-use one-pot amplification method, where results could be examined and visualized within 45 min without an expensive thermal cycler and other sophisticated lab equipment and conditions. The standardized LAMP reaction was validated using 100 well characterized *P. falciparum* samples comprising 40 mutants (S108N) and 60 wild-type strains. In all experiments, the mutant and control samples were used to check the precision of HNB–LAMP analysis, validated by DNA sequencing (Table 2). The results identified by HNB–LAMP were in full agreement with the results of gold standard sequencing analysis (Table 2 and Appendix A), which supports that the standardized HNB–LAMP approach is highly sensitive (7 parasites/µL of genomic DNA) and reliable to detect PYR-resistant (S108N) mutants and its applicability to detect at low parasitemia. This easy-to-use approach can be used for the rapid monitoring of the PYR-resistant mutants of *P. falciparum* in resource limiting settings.

## 4. Discussion

The LAMP assay has been extensively applied in nucleic acid analysis due to its ease of use, high speed, efficiency, and specificity. In recent years, this assay has been successfully applied for the diagnosis of pathogenic microbes and drug resistant mutant detection through nucleic acid amplification [1,18,19]. Moreover, the application of the LAMP approach for antimalarial resistance detection was initially reported in our previous studies on chloroquine-resistance, and another study by Yongkiettrakul et al (2016) has described pyrimethamine resistance detection with the lateral flow dipstick method [14,15,16]. Due to the fast emergence of pyrimethamine (PYR)-resistance in malaria-endemic areas, it is becoming a challenge to control malaria. Regular monitoring of the mutation status of PYR as a partner drug is essential to reframe anti-malarial policy. To overcome this limitation, for effective treatment, we have developed a novel HNB–LAMP assay for the rapid detection of the pyrimethamine-resistant mutants of *P. falciparum*. 

For an accurate LAMP method, it is always challenging to design highly specific primers and optimize the reaction conditions for the one-pot amplification of target DNA. Considering these points, HNB–LAMP primers were designed and synthesized, adding one mismatched nucleotide at different positions at the annealing point of FIP. We selected the most appropriate primer set (FIPM1) to detect the S108N mutation in all samples (Figure 1 and Table 1). Furthermore, the accuracy of the LAMP assay was also confirmed in mixed strains of plasmodium having heterogeneous parasite populations; no cross reactivity was observed (data not shown). Correspondingly, in the mixed population of mutant and wild type strains, the assay showed reliable positive detection. 

Following this primer set (FIPM1), we optimized the selected concentration of reaction components and reaction conditions of HNB–LAMP to reduce the detection time. The LAMP reaction was optimized at a temperature range of 57–66 °C and examined at every 15 min time interval. The standardized HNB–LAMP assay can detect mutations with a 45 min reaction time within a temperature range of 60–66 °C by visualization.

The LAMP reaction mechanism is based on the binding of pyrophosphate ions with metal indicators, resulted in a white precipitate. Consequently, hydroxynaphthol blue (HNB) can be used as a metal ion in the LAMP reaction and the results can be visualization by observing color change [20], CuSO4 [21], or calcein [14]. Additionally, LAMP reactions can also be visualized with the naked eye by using DNA intercalating dyes like SYBR green [22] or Picogreen [23]. These dyes can be added after a reaction is completed, which further increase the rate of contamination in the reaction due to the exposed operation [11,24]. To overcome the limitations of contamination during visual analysis, we have demonstrated here a robust and rapid alternative approach for visual detection using indicator dye HNB, which is low-priced and simple to use and exhibit. Other visual detection methods for LAMP target amplification use the color transformation of a metal sensitive-indicator—such as calcein, which causes a shift from bright yellow to yellow, and hydroxylnaphthol blue, indicated by a dark blue to blue shift [15]—for real-time visual detection. This study favors the continued improvement of LAMP analysis with the use of hydroxynaphthol blue indicator for the clear visual detection of the amplification product as reported by Goto et al. [20]. It takes advantage of LAMP in facilitating field, point of care, and even basic rapid laboratory detection.

Traditional detection methods of the pyrimethamine-resistant mutants of *P. falciparum* are limited by the requirements of expensive equipment, insufficient specificity, and rather low amplification efficiency. Thus, it is indeed required to develop a rapid, economic, and high-throughput approach for detecting PYR resistance in *P. falciparum*. This HNB based SNP–LAMP is optimized and established as being simple, more rapid, and more feasible to detect the S108N mutation with improved efficiency and a shortened detection time. To confirm the redundancy, this HNB–LAMP assay was tested following known S108N mutant samples from different regions of India. Meanwhile, the sensitivity of the HNB–LAMP approach was evaluated through the tenfold serial dilution (10^−1^ to 10^−6^) of genomic DNA, which showed a detection limit of 10^−4^-fold dilution, accounting for 7 parasites/μL, supporting the high sensitivity (Figure 7). The results represent a good indicator of its applicability for detecting a low parasite burden.

The LAMP approach has been used by Abdul Ghani et al. [1] as a point of care test (POCT) for the diagnosis of malaria following the other studies on malaria diagnosis [25,26,27,28], even though there are few reports for the evaluation of antimalarial drug resistance using LAMP, like some reports on bacteria [29,30]. A previous study on drug resistance was reported by Yongkiettrakul et al (2017), based on the mutant (N51I) associated with the *Pfdhfr* gene for pyrimethamine resistance detection in *P. falciparum* using the LAMP assay [16]. The above study had advantage to use directly lysed blood samples butit relied on a special primer designing scheme. On the contrary, however, the present study showed the superior primer targets S108N mutation rather than additive mutant N51I. Moreover, we evaluated the HNB LAMP assay with a larger pool of malarial isolates, included the testing of mix parasite population, which indicated 100% accuracy in discriminating the mutant and wild type strains, even in heterogeneous populations, and consequences showed reliable positive detection.

This novel HNB–LAMP assay is highly sensitive (7 parasites/μL) and specific for the discrimination of the pyrimethamine-resistant mutant strains of *P. falciparum* and have advantages over traditional PCR-based molecular methods. However, one of the major limitations of this assay compared to PCR is the complexity of designing LAMP primers. LAMP requires the design of a set of at least two primer pairs to identify six regions of the target gene to increase the efficiency of the reaction, with the possibility of using two additional loop primers to accelerate the reaction [11,12]. 

In conclusion, the newly developed HNB–LAMP assay can be used for the early warning of resistance risk to the partner drug pyrimethamine in resource-limited settings, and can be used for surveillance of the S108 mutation in the endemic areas as well. In the future, we believe this assay can serve as a surveillance tool and guide a treatment algorithm for the partner drug pyrimethamine in a clinically significant time frame, prevent unnecessary use of supplementary drugs that may drive additional resistance, and avoid lengthier treatment regimens that cause toxicity for the patient. However, more work is required to further evaluate the clinical utility of this test prospectively in SP endemic areas. 

## Figures and Tables

**Figure 1 ijerph-16-01635-f001:**
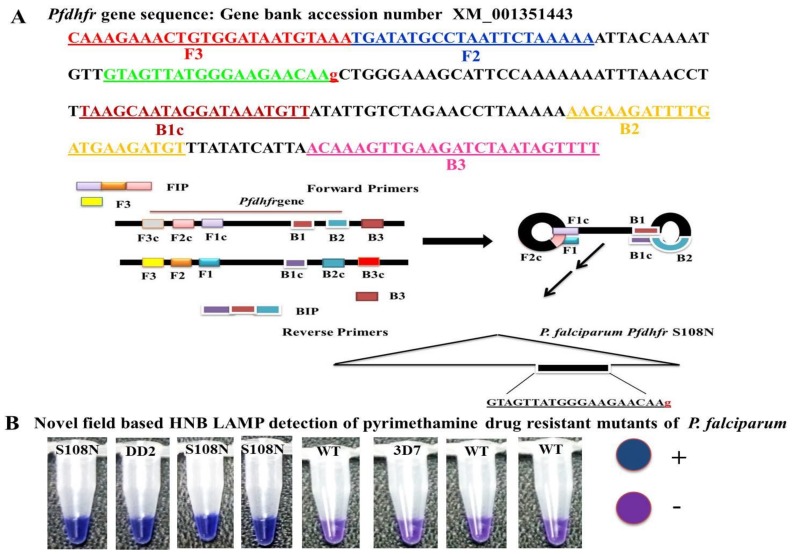
Sequence alignment of *Pfdhfr* gene and diagrammatic representation of loop primers used in the study and simple hydroxynaphthol blue (HNB) based single nucleotide polymorphism loop-mediated isothermal amplification (SNP–LAMP) detection of S108N mutants of *P. falciparum*. (**A**) Nucleotide sequence alignment of the *Pfdhfr* gene target region: Sequences of *Pfdhfr* gene primers F3, B3, FIP, and BIP used in the SNP–LAMP assay. Single point mutation shown by red small letter (S108N AGC→AAC) inserted in forward inner primer (FIP) allied with pyrimethamine-resistant *P. falciparum* malaria. Sites of primer binding region in the reference sequence (Pf; Gene bank accession no. XM_001351443) was indicated by underlines. Schematic representation of the *Pfdhfr* gene loop primers: Edifice of the inner primers FIP (F1c + F2) and backward inner primer (BIP) (B1c + B2) were shown. F1c and B1c are complementary to F1 and B1. (**B**) Novel SNP–LAMP assay based on HNB indicator for visual detection of pyrimethamine-resistant mutants (S108N) in *P. falciparum*. A platform toward development of easy to use visual SNP–LAMP assay for fast detection of antimalarial drug resistant mutants in *P. falciparum*.

**Figure 2 ijerph-16-01635-f002:**
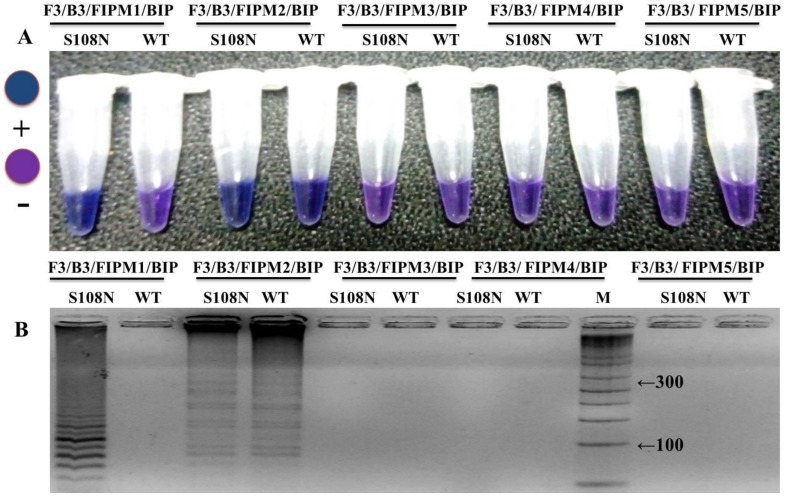
Optimization of in-house designed LAMP primers. (**A**) Assessment was based on the hydroxynaphthol blue (HNB) visualization of the color change in the tube. (**B**) A ladder like pattern was on the gel electrophoresis analysis of the LAMP products for mutant detection.

**Figure 3 ijerph-16-01635-f003:**
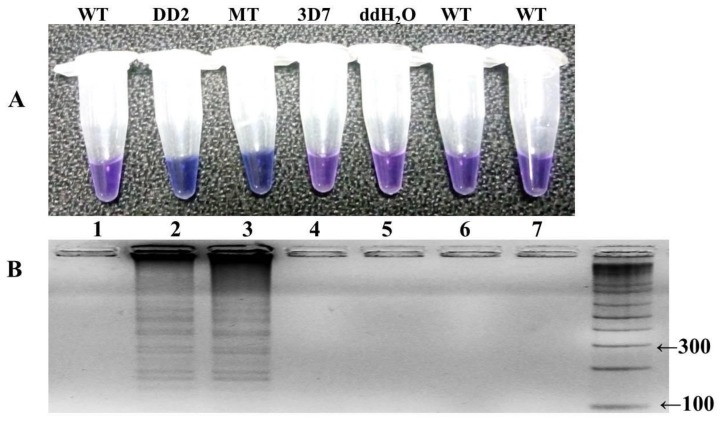
HNB–LAMP assay detection of the S108N mutation with the primer set FIPM1 in pyrimethamine (PYR)-resistant samples of *P. falciparum* and the assessment of LAMP products on gel electrophoresis. (**A**) PYR-resistant mutant strains of *P. falciparum* are indicated by the blue color shown in tube no. 3, including positive control in tube no. 2, while, the negative shows a violet color in tubes no. 1, 6, and 7, including negative control 3D7 (tube no. 4). (**B**) Taxation was based on gel electrophoresis: lane 2 and 3 showed amplification (ladder-like pattern) for PYR-resistant mutants with positive control DD2 (lane 2); and no band pattern in lanes 1, 4, 6, and 7 for wild type and negative strains.

**Figure 4 ijerph-16-01635-f004:**
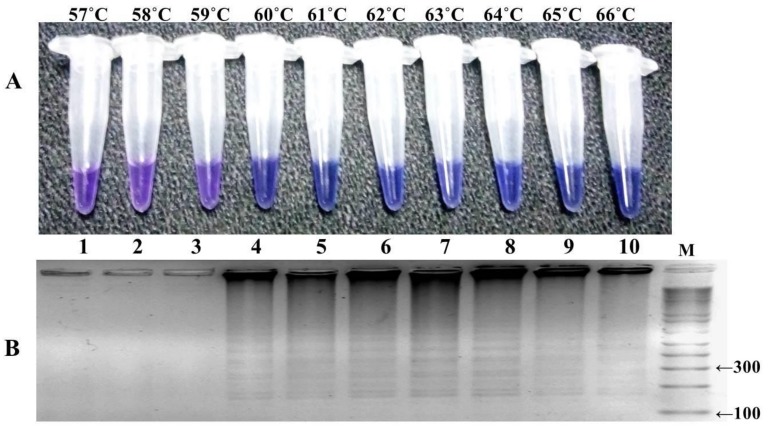
Optimization of temperature conditions for HNB–LAMP. (**A**) HNB color change visualization at different temperatures: there was no color change at 57 °C to 59 °C, which indicated in tube no. 1, 2, and 3 that the reaction was not amplified at this range, which was re-confirmed on 3% agarose gel as shown in lanes 1, 2, and 3. (**B**) Assessment was based on the gel electrophoresis analysis of the LAMP products: M-100 bp ladder, lanes 1 to 10 show the tested temperature range of 57–66 °C, and the maximum amplification (ladder-like pattern) was observed at 60–66 °C as indicated in lanes 4 to 10.

**Figure 5 ijerph-16-01635-f005:**
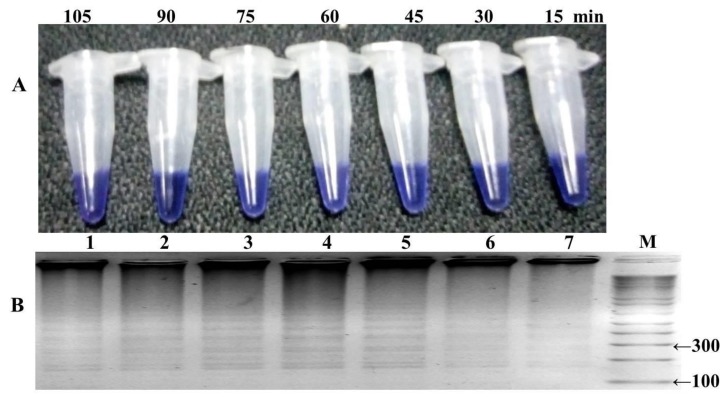
Optimization of the HNB–LAMP reaction at time intervals. (**A**) Assessment was based on HNB color change visualization. A positive reaction started to color change from violet to blue after 45 min at 60–66 °C, indicated by tube no. 5. (**B**) Validation through gel electrophoresis at different time intervals: M-100 bp ladder, lanes 1 to 7 showed a time range from 15 to 105 min, and the maximum amplification of the LAMP product in a short time was observed at 45 min (lane 5) as compared to the amplification efficiency within 30 and 15 min indicated in lanes 6 and 7.

**Figure 6 ijerph-16-01635-f006:**
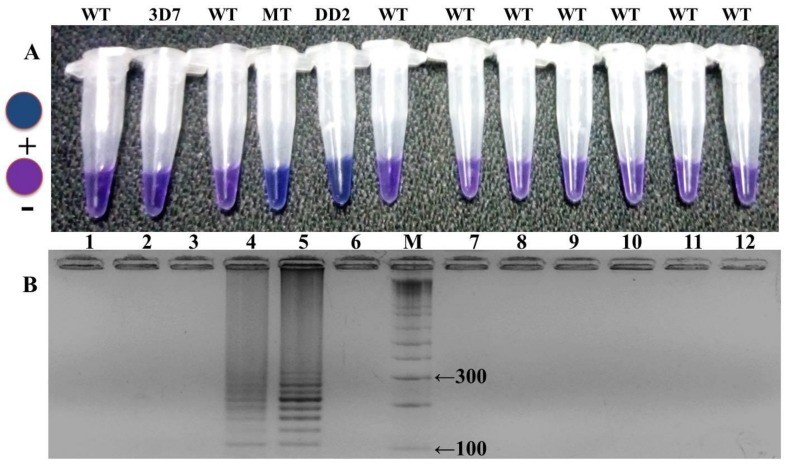
Accuracy and specificity of HNB–LAMP assay. (**A**) Assessment was based on the HNB color change visualization of the LAMP product. *P. falciparum* mutants distinguished by blue color change in reaction as shown in tubes no. 4 and 5 with the positive control DD2 indicated in tube no. 5; in disparity, the PYR wild-type strains showed a violet color reaction indicated by tubes no. 1, 2, 3, 6, 7, 8, 9, 10, 11, and 12 with wild-type control 3D7 (tube no. 2). (**B**) Assessment was based on gel electrophoresis: M-100 bp ladder, lanes 4 and 5 showed amplification (ladder-like pattern) in PYR-resistant mutants with positive control DD2 (lane 5), while no amplification was observed in wild type samples in lanes 1, 2, 3, 6, 7, 8, 9, 10, 11, and 12, including the wild-type control 3D7.

**Figure 7 ijerph-16-01635-f007:**
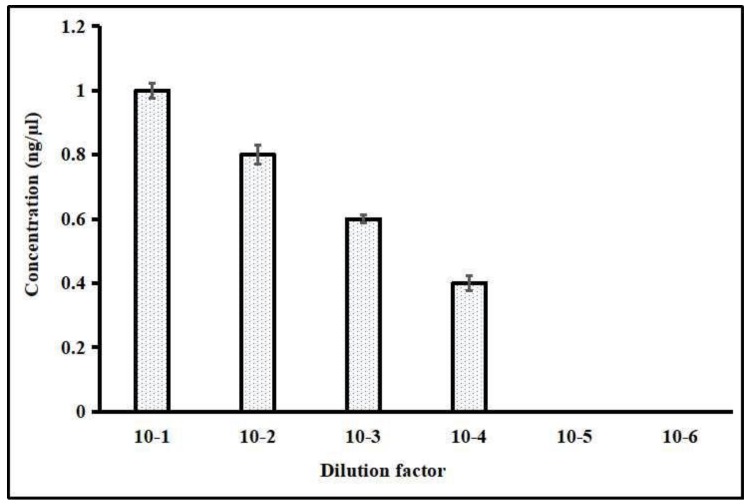
Graphical representation for sensitivity of LAMP. The lowest dilution limit of HNB–LAMP detection was up to 10^−4^ (7 parasites/μL).

**Table 1 ijerph-16-01635-t001:** In-house designed LAMP primers of *Pfdhfr* gene targeting S108N codon.

*Pfdhfr* Gene Sequence of in-House Designed LAMP Primer Target S108N (5’-3’) ^a^	Usage
F3: 5′-CAAAGAAACTGTGGATAATGTAA-3′	Outer forward primer
B3: 5′-AAAACTATTAGATCTTCAACTTTGT-3′	Outer backward primer
BIP: 5′-TAA……………………………………………………..CTT-3′ *	Inner backward primer
FIPM1: 5′-TATGTTCTT……………………………………………….TAAAAA-3′ *	Inner forward primer To distinguish the pyrimethamine-resistant mutants (S108N) of *P. falciparum*
FIPM2: 5′-TTAGT………………………………………………TTCTAAAAA-3′ *	
FIPM3: 5′-ACTTTG…………………………………………………….AAAAA-3′ *	
FIPM4: 5′-CTAT……………………………………………………..CTAAAAA-3′ *	
FIPM5: 5′-CTTG…………………………………………………….TCTAAAAA-3′ *	

^a^ Nucleotides in ends are altered from the *Pfdhfr* gene sequence of *P. falciparum*. Nucleotides indicated by red color are mismatches inserted manually to discriminate pyrimethamine-resistant mutants (S108N) from those of the wild type. * Patent pending.

**Table 2 ijerph-16-01635-t002:** Sensitivity, specificity, and accuracy of the HNB–LAMP detection method.

Molecular Methods	95% Confidence Interval		Accuracy	Measure of Agreement (κ)
	Sensitivity	Specificity		
HNB–LAMP–gel	1	1	100%	1
HNB–LAMP–visible	1	1	100%	1
DNA Sequencing	1	1	100%	1
PCR–RFLP	1	0.96	98%	0.96

Kappa coefficient (κ) statistics range (−1 to + 1), *p* < 0.0005 measures the observed percentage of agreement between tests. HNB–LAMP–gel, loop-mediated isothermal amplification by gel electrophoresis; HNB–LAMP–visible, loop-mediated isothermal amplification visualized by naked eye or UV light; RFLP, restriction fragment length polymorphism.

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
