# Peer review of "Development and Evaluation of a Novel HNB Based Isothermal Amplification Assay for Fast Detection of Pyrimethamine Resistance (S108N) in Plasmodium falciparum"

_ijerph, 2019, doi:10.3390/ijerph16091635_

Round 1

Reviewer 1 Report

Development and evaluation of a novel HNB based isothermal amplification assay for fast detection of Pyrimethamine resistance (S108N) in Plasmodium

Chahar et al

Rapid detection methods for the presence of Plasmodium species such as optimized LAMP might be an important diagnosis tool for malaria (Towards rapid genotyping of resistant malaria parasites: could loop-mediated isothermal amplification be the solution;Abdul-Ghani R1, 2014). Several of such assays have been described and the authors recently published a paper describing a primerset for the detection of Chloroquine resistant Plasmodium species. The same group here establishes a LAMP assay to detect Pyrimethamine resistant Plasmodium species.

The authors suggest in the discussion that “this assay can serve as a surveillance tool and guide treatment algorithm for partner drug pyrimethamine in a clinically significant time frame, prevent unnecessary use of supplementary drugs that may drive additional resistance and avoid lengthier treatment regimens that cause toxicity” This might be a rather strong claim given the fact that no field research is performed and the rather large prevalence of this mutation on large parts of the world, (see e.g. Prevalence of Plasmodium falciparum Molecular Markers of Antimalarial Drug Resistance in a Residual Malaria Focus Area in Sabah, Malaysia.Norahmad et al 2016).

Nevertheless this development can be of added value for clinical detection and could be used as treatment guide if more independently validated genotype detection assays were to be compared.

The paper describes the development of a primerset for LAMP detection of the S108N plasmodium species

·         They confirm that it detects from a reference set with astonishing? 100% accuracy, since the result is not presented in a figure but in a supplement table. I prefer not to evaluate to keep myself anonymous, I’m not able to evaluate this result.

·         The detection limit for this assay was presented as 10-2 and 10-4 (in respectively material and methods and abstract and result), this should be clarified. This is further specified as 2ng/ul, 2 ng of genomic DNA? What is the Plasmodium load and how thus this relate to real patient samples. Most importantly, is this detection limit good enough for field applications, similar methods detect fg of DNA, is this detection limit low enough to detect the presence of this mutation in a patient : A more quantitative method should be used to quantify the minimum number of copies that could be detected with this method, and this should be correlated with the amount of plasmodium expected in patients

·         The meaning of the Y-scale, 0-1.ng/ul DNA of figure 6 is unclear to me, to me the amount of genomic DNA is not so relevant but rather the amount of Plasmodium DNA that was present

Author Response

Response to Reviewer #1 remarks

Remark 1 #   

 They confirm that it detects from a reference set with astonishing? 100% accuracy, since the result is not presented in a figure but in a supplement table. I prefer not to evaluate to keep myself anonymous, I’m not able to evaluate this result.

Answer 1

For the confirmation of LAMP specificity with 100% accuracy we check the cross reactivity with a mixture of heterogeneous parasite populations. Pyrimethamine-resistant parasites (DD2) were mixed with pyrimethamine-sensitive parasite (3D7). DNA samples of mixed parasite populations were mixed with DD2 and 3D7, respectively, were detected by LAMP assay using Pfdhfr gene S108N as a reliable marker to detect PYR resistant mutants of P. falciparum and the experiment was repeated three times with positive and negative control reference strains. LAMP assay clearly discriminate the mutant and wild type strains from genetically mixed population. The results were further validated by RFLP and Sequencing (data not given).

Remark 2 #  

The detection limit for this assay was presented as 10-2 and 10-4 (in respectively material and methods and abstract and result), this should be clarified. This is further specified as 2ng/ul, 2 ng of genomic DNA? What is the Plasmodium load and how thus this relates to real patient samples? Most importantly, is this detection limit good enough for field applications, similar methods detect fg of DNA, is this detection limit low enough to detect the presence of this mutation in a patient: A more quantitative method should be used to quantify the minimum number of copies that could be detected with this method, and this should be correlated with the amount of plasmodium expected in patients.

Answer 2.

To evaluate the limit of LAMP detection, the plasmodium load in P. falciparum samples was calculated by following microscopic verification; The parasite samples chosen for these dilutions contained initial densities ranging between 0.02 and 1 parasitized cells per 100 red blood cells (referred to as 0.02% and 1% parasitemia, respectively), as determined by microscopic examination. On the basis of parasitemia, we evaluated the limit of parasite detection by using serial dilutions of genomic DNA samples having range of low to high parasitemia. Genomic DNA concentration was quantified by using spectrophotometer (Nanodrop Technologies), where a tenfold dilutions were used to perform the reaction. The limit of LAMP detection was observed up to104 fold dilution having < 2 ng/μl concentration with low parasite load (7 parasite/ μl).    

Remark 3 #  

The meaning of the Y-scale, 0-1.ng/ul DNA of figure 6 is unclear to me, to me the amount of genomic DNA is not so relevant but rather the amount of Plasmodium DNA that was present

Answer 3

We agree with the reviewer positive comment; the genomic DNA has a mixture of human DNA and plasmodium DNA. In this study we quantified genomic DNA concentration to observed the quantitative detection limit of LAMP assay, but the main emphasize in this study was to observed the detection limits at low parasitemia and the plasmodium load was calculated by following microscopic verification;

Reviewer 2 Report

Dear Editor,

I have carefully read the submission of Chahar and colleagues entitled ‘Development and evaluation of a novel HNB based isothermal amplification assay for fast detection of Pyrimethamine resistance (S108N) in Plasmodium falciparum’.

This submission is a technical paper presenting the development of a LAMP based approach for the quick and cheap identification of P. falciparum strains that are Pyrimethamine-resistant.

To address this, the authors developed and evaluated sets of LAMP primers and optimized for different temperatures, times, and samples’ DNA concentration.

Overall, the study is well written and performed, with no severe drawbacks. However, I have some concerns (raised below) that I believe that, if addressed by the authors, will make the submission more robust and easily followed by a broader audience.

Concerns:

1.      Minor English editing: see for example:

-          ‘Pyrimethamine and sulphadoxine inhibit(s)’.

-          ‘However (better ‘moreover’), their use is limited’.

-          artemisisnin’ (correct this).

-          ‘to determine the minimum detected limit’ (correct to ‘detection limit’).

-          ‘only one primer set (FIPM1) was found to be most appropriate’ (sentence needs editing).

-          ‘Our consequences indicated’ (the use of ‘consequences’ does not seem appropriate on many occasions. Please check and revise.

-          ‘gold-standard sequencing consequences’ I am not sure about the meaning of this sentence.

-          …and some others throughout the manuscript.

2.      M+M section - ‘2.2. P. falciparum Samples’: a better description of the samples used is needed, especially for non-experts in the field. What are the ‘samples’ exactly?

3.      M+M section – ‘The LAMP primer sets and their sequence alignment used in this study are given in Table S1 and Figure S1’: I believe this information is too vital to be only in the supplementary material. I would suggest Figure S1 and a list of the primers designed to be transferred in the main article. On the other hand, the more extended supplementary material can also be presented ‘as-is’.

4.      M+M section – ‘through a commercially available sequence service’: specify the sequencing service.

5.      M+M – ‘2.8. Statistical Analysis’: this needs extensive revision. Using SPSS does not say much. Please specify here all the statistical analysis performed.

6.      Please check the right order of Figures. Figure 2 is after 3 and 4.

7.      Major concern in M+M, Results, and Discussion: the authors present their methodology using ‘clean’ and well-characterized samples. However, it is not clear how this methodology will be applied in the field. What kind of samples will be collected and analyzed? The authors should discuss this, which is a clear limitation since many times diagnostics do not work properly when they must be really applied, especially under the conditions and for the purposes the authors want to use them for.

8.       Major concern in M+M, Results, and Discussion: I may be wrong, but it seems to me that it is important not only to distinguish between resistant and sensitive strains but also for the presence or not of Plasmodium in general. I think that the authors must shed some light on how this approach will be used or if it needs to be combined with something else. Again, the authors have to clearly present all the limitations of their approach. 

Author Response

Response to reviewer #2 remarks

Remark 1

1.      Minor English editing: see for example:

-          ‘Pyrimethamine and sulphadoxine inhibit(s)’.

-          ‘However (better ‘moreover’), their use is limited’.

-          ‘artemisisnin’ (correct this).

-          ‘to determine the minimum detected limit’ (correct to ‘detection limit’).

-          ‘only one primer set (FIPM1) was found to be most appropriate’ (sentence needs editing).

-          ‘Our consequences indicated’ (the use of ‘consequences’ does not seem appropriate on many occasions. Please check and revise.

-          ‘gold-standard sequencing consequences’ I am not sure about the meaning of this sentence.

-          …and some others throughout the manuscript.

Answer 1

As per reviewers suggestion minor English editing is done throughout the manuscript highlighted yellow.

Remark 2

2.      M+M section - ‘2.2. P. falciparum Samples’: a better description of the samples used is needed, especially for non-experts in the field. What are the ‘samples’ exactly?

Answer 2

According to reviewer suggestion the detailed description of samples is now added in the revised manuscript which is highlighted yellow.

2.2. P. falciparum Samples

Forty  P. falciparum positive samples containing the Pfdhfr (N108) mutation and sixty samples comprising a Pfdhfr wild type allele (S108) were obtained from the Ranchi & Jharkhand, India. These samples were previously characterized by PCR-RFLP for mutation (S108N) associated with pyrimethamine resistance in P. falciparum and were used to determine the efficacy of the HNB-LAMP assay. The control strains, including a wild-type control (3D7) and mutant control (DD2) were provided by the Malaria Parasite Bank of Host Institute. Clinical sample collections were approved by the Institutional Ethical Review Committee of the National Institute of Malaria Research (NIMR), New Delhi, India.

Remark 3

3.      M+M section – ‘The LAMP primer sets and their sequence alignment used in this study are given in Table S1 and Figure S1’: I believe this information is too vital to be only in the supplementary material. I would suggest Figure S1 and a list of the primers designed to be transferred in the main article. On the other hand, the more extended supplementary material can also be presented ‘as-is’.

Answer 3

 As per reviewer’s suggestion the information regarding Table S1 and Figure S1’ have been shifted from the supplementary material and now included in the main manuscript highlighted yellow.

Remark 4

4.      M+M section – ‘through a commercially available sequence service’: specify the sequencing service.

Answer 4

According to reviewer’s suggestion detail of sequence service has been specified in the revised manuscript highlighted yellow.

2.7. Corroboration of LAMP Assay via Sequencing

For the endorsement and precision of LAMP results, PYR-resistant samples with Pfdhfr S108N mutation and all the wild types were validated through DNA sequencing. The specific target of Pfdhfr gene from all samples were amplified through forward (F3) and backward (B3) primers (Table 1) using PCR and the PCR products were sequenced through a commercially available sequence service of New England Biolabs, USA.

Remark 5

5.      M+M – ‘2.8. Statistical Analysis’: this needs extensive revision. Using SPSS does not say much. Please specify here all the statistical analysis performed

Answer 5

As per reviewer’s suggestion the information regarding Statistical Analysis has been specified and the description is now added in the revised manuscript highlighted yellow.

2.8. Statistical Analysis

Statistical analysis was accomplished by using the Statistical Package for the Social Sciences (SPSS) software version 10. The frequencies and 95% confidence interval (CI) were used for categorical variables. The sensitivity, specificity and accuracy of LAMP, PCR-RFLP and gold-standard sequencing reference test were evaluated by Chi square test or Fisher’s exact test. The level of statistical significance was set at a value of p0.05. In this study, each test was repeated thrice, leading to three practical modalities.

Remark 6

6.      Please check the right order of Figures. Figure 2 is after 3 and 4.

Answer 6

As per reviewer’s suggestion the information regarding correct order of figures has been updated in the revised manuscript highlighted yellow.

Remark 7

7.      Major concern in M+M, Results, and Discussion: the authors present their methodology using ‘clean’ and well-characterized samples. However, it is not clear how this methodology will be applied in the field. What kind of samples will be collected and analyzed? The authors should discuss this, which is a clear limitation since many times diagnostics do not work properly when they must be really applied, especially under the conditions and for the purposes the authors want to use them for.

Answer 7

Reviewer raised very interesting question. In fact we had also optimized the LAMP assay to detect PYR resistance in genetically mixed parasite populations.  For the confirmation of LAMP specificity we check the cross reactivity with a mixture of heterogeneous parasite populations. Pyrimethamine-resistant parasites (DD2) were mixed with pyrimethamine-sensitive parasite (3D7). DNA samples of mixed parasite populations were mixed with DD2 and 3D7, respectively, were detected by LAMP assay using Pfdhfr gene S108N as a reliable marker to detect PYR resistant P. falciparum and the experiment was repeated three times with positive and negative control reference strains. LAMP assay clearly discriminate the mutant and wild type strains from genetically mixed population. The results were further validated by RFLP and Sequencing (data not given). This information is now updated in the revised manuscript highlighted yellow.

Remark 8

8.       Major concern in M+M, Results, and Discussion: I may be wrong, but it seems to me that it is important not only to distinguish between resistant and sensitive strains but also for the presence or not of Plasmodium in general. I think that the authors must shed some light on how this approach will be used or if it needs to be combined with something else. Again, the authors have to clearly present all the limitations of their approach.

Answer 8

 There is no study available with the combination of plasmodium species and resistance detection because the major limitation is the complexity to design the loop primers (Veigas et al 2013). In this study we used the in-house designed LAMP primers specific to discriminate pyrimethamine resistant mutants and wild type strains of Plasmodium falciparum but not for all species of plasmodium. According to the reviewers suggestion the limitations of LAMP assay is now added in the revised manuscript highlighted yellow.

Reviewer 3 Report

Artemisinin is currently the most effective antimalarial drug available, yet resistant parasites have been identified in South-East Asia.  To limit the spread of Artemisinin resistance monotherapy is actively discouraged, instead combination therapies that include co-treatment with other antimalarial drugs are preferred.  However, resistance to potential partnering drugs is widespread; before deciding on an Artemisinin combination therapy a doctor would ideally have a cheap and rapid test that allowed pre-treatment screening to identify whether parasites are present that are resistant to one component of the therapy. 

Sulphadoxine-pyrimethamine are long-lasting inhibitors of P. falciparum dihydrofolate reductase thymidylate synthase and are often included in Artemisinin combination therapies.  P. falciparum resistance to Sulphadoxine-pyrimethamine is widespread (Okell et al.2017) and involves a key Dhfr mutation (S108N) often co-occuring with the N51I and C59R additive mutants. Chahar et al.describe a potential test for the S108N Pfdhfr mutation that is rapid and requires no specialised equipment.  The work reports a primer set that yields S108N-specific DNA amplification via single nucleotide polymorphism loop-mediated isothermal amplification (SNP-LAMP) assay.  The authors used a similar assay to detect the chloroquine-resistance associated with the Pfcrt K76T mutant in earlier work (Chahar et al.Exp parasitology 2018); while other groups have reported SNP-LAMP primer sets for artemisinin resistance (Mohon et al. 2018 Open Forum Infect Disease) but also previously for pyrimethamine resistance using a primer set based around the co-occuring Dhfr N51I mutation (Yongkiettrakul et al. 2017 parasitology international.)

Comment on prior work: 

The fact that this Yongkiettrakul et al. paper is not referenced in this work appears to be something of an oversight.  This manuscript has been available online since late 2016. In the introduction and again in the discussion the authors state the following about their 2017 Scientific reports paper:

“However, the application of LAMP in the field of antimalarial resistance detection had not been reported until the detection of chloroquine- resistance in P. falciparum was revealed by our group [14].” 

This seems unfair on the work of Yongkiettrakul et al, especially so given the similarities between the current paper and that groups parasitology international paper entitled Simple detection of single nucleotide polymorphism in Plasmodium falciparum by SNP-LAMP assay combined with lateral flow dipstick”  Given the similarities between the two reports Chahar et al.would ideally comment on the comparisions with this prior work. The primers of  Chahar et al. appear somewhat superior as they target the key S108N mutation rather than additive mutant N51I and they have applied their test to a larger pool of malarial isolates.  But the earlier primer set and methodology of Yongkiettrakul et al.appears to yield similar selectivity, is proven to do so from lysed blood samples instead of requiring DNA extraction and their lateral flow dipstick detection methods appears, to my eyes, to be better suited to clinical use than the purple to violet colour change afforded by the authors use of HNB.  

Codon specificity:

The S108N mutant requires a single nucleotide change from the wildtype sequence but asparganine has two potential codons (AAC and AAT).  The authors SNP-LAMP reaction primer set appears to give good specificity for the Asn codon AAC.  Given they have sequence information for a number of malarial isolates do they have any evidence that the codon AAT is ever used by the parasite and if so whether their LAMP primer set reacts against this sequence?

Minor Gramatical changes:

“Page 1

“..not bind to the active site of the enzyme and result in resistance for pyrimethamine and sulphadoxine “

Page 2

“…and it would be cause of treatment failure in the near future.” 

Page 3

“The LAMP assay was performed..”

Page 7

“..The primer set FIPM1 was finally selected for the further study..” 

Page 9

The LAMP constancy and repeatability was confirmed withusing 40 mutant (S108) samples and 60 wild-type samples including control strains

Each reaction was done thrice autonomously for LAMP assay

“…where results could be examined and visualized..”

“..and it’s applicability to detect at low parasitemia.” 

To the best of our knowledge, this is the first report wherein LAMP has been applied for the fast and easy detection of partner drug (PYR) resistance in P. falciparum malaria.” 

            àsee Yongkiettrakul et al 2017

Page 10:

            “..after a reaction is completed but their final edition and the exposed operation increase..” 

     (reword sentence)

Author Response

Response to reviewer #3 remarks

Remark 1

1.      Comment on prior work: 

The fact that this Yongkiettrakul et al. paper is not referenced in this work appears to be something of an oversight.  This manuscript has been available online since late 2016. In the introduction and again in the discussion the authors state the following about their 2017 Scientific reports paper:

“However, the application of LAMP in the field of antimalarial resistance detection had not been reported until the detection of chloroquine- resistance in P. falciparum was revealed by our group [14].” 

This seems unfair on the work of Yongkiettrakul et al, especially so given the similarities between the current paper and that groups parasitology international paper entitled Simple detection of single nucleotide polymorphism in Plasmodium falciparum by SNP-LAMP assay combined with lateral flow dipstick”  Given the similarities between the two reports Chahar et al.would ideally comment on the comparisions with this prior work. The primers of  Chahar et al. appear somewhat superior as they target the key S108N mutation rather than additive mutant N51I and they have applied their test to a larger pool of malarial isolates.  But the earlier primer set and methodology of Yongkiettrakul et al.appears to yield similar selectivity, is proven to do so from lysed blood samples instead of requiring DNA extraction and their lateral flow dipstick detection methods appears, to my eyes, to be better suited to clinical use than the purple to violet colour change afforded by the authors use of HNB. 

Answer 1

According to reviewer’s suggestions the Yongkiettrakul et al, 2017 reference is now updated in the revised manuscript and the highlights of this study is added in the introduction and  discussion section highlighted yellow.

As per reviewer’s suggestions these lines (given below) are now deleted from the manuscript.

 “However, the application of LAMP in the field of antimalarial resistance detection had not been reported until the detection of chloroquine- resistance in P. falciparum was revealed by our group [14].” 

Remark 2

2.      Codon specificity:

The S108N mutant requires a single nucleotide change from the wildtype sequence but asparganine has two potential codons (AAC and AAT).  The authors SNP-LAMP reaction primer set appears to give good specificity for the Asn codon AAC.  Given they have sequence information for a number of malarial isolates do they have any evidence that the codon AAT is ever used by the parasite and if so whether their LAMP primer set reacts against this sequence?

Answer 2

In this study we designed and tested different sets of LAMP primers in each set we have replacements with first and second base change in the forward inner primer (FIP) to discriminate mutation at 108 codon of Pfdhfr gene. LAMP and DNA sequencing analysis showed that we detected both replacements AAC and AAT with the FIPM1 primer set but in total only 2 samples showed AAT replacement.

Remark 3

Minor Gramatical changes:

“Page 1

“..not bind to the active site of the enzyme and result in resistance for pyrimethamine and sulphadoxine “

Page 2

“…and it would be cause of treatment failure in the near future.” 

Page 3

“The LAMP assay was performed..”

Page 7

“..The primer set FIPM1 was finally selected for the further study..” 

Page 9

“The LAMP constancy and repeatability was confirmed withusing 40 mutant (S108) samples and 60 wild-type samples including control strains“

“Each reaction was done thrice autonomously for LAMP assay“

“…where results could be examined and visualized..”

“..and it’s applicability to detect at low parasitemia.” 

“To the best of our knowledge, this is the first report wherein LAMP has been applied for the fast and easy detection of partner drug (PYR) resistance in P. falciparum malaria.” 

            àsee Yongkiettrakul et al 2017

Page 10:

            “..after a reaction is completed but their final edition and the exposed operation increase..” 

     (reword sentence)

Answer 3

As per reviewers suggestion minor grammatical changes is done in pages (1, 2, 3, 7 and 9) throughout the manuscript and the sentence is reword at page 10 highlighted yellow.

According to reviewers suggestion the lines given below is now deleted from the manuscript.

“To the best of our knowledge, this is the first report wherein LAMP has been applied for the fast and easy detection of partner drug (PYR) resistance in P. falciparum malaria.” 
